# The Levels of Serum Serotonin Can Be Related to Skin and Pulmonary Manifestations of Systemic Sclerosis

**DOI:** 10.3390/medicina58020161

**Published:** 2022-01-21

**Authors:** Marin Petrić, Dijana Perković, Ivona Božić, Daniela Marasović Krstulović, Dušanka Martinović Kaliterna

**Affiliations:** Department of Rheumatology and Clinical Immunology, University Hospital of Split, Šoltanska 1, 21000 Split, Croatia; dijana.perkovic@hotmail.com (D.P.); ivona.bozic.7@gmail.com (I.B.); daniela.marakrst@gmail.com (D.M.K.); martikalit@gmail.com (D.M.K.)

**Keywords:** fibrosis, modified Rodnan skin score, pulmonary function test, serotonin, skin, systemic sclerosis

## Abstract

*Background and Objective*: The most prominent feature of systemic sclerosis (SSc), besides vasculopathy and autoimmune disorders, is excessive fibrosis. Serotonin affects hemostasis and can induce vasoconstriction, which is presumed to be one of the pathophysiological patterns in SSc that leads to fibrosis. Our aim was to explore the possible association of serotonin with some of the clinical features of SSc in our cohort of patients. *Materials and Methods*: We measured serotonin levels in sera of 29 female SSc patients. Patients were 41–79 years old, their average disease duration was 9 years. Serotonin values were analyzed in correlation with clinical and laboratory parameters, such as modified Rodnan skin score (mRSS), digital ulcers (DU), and spirometry parameters-forced expiratory volume in the first second (FEV1), forced vital capacity (FVC), and lung diffusion capacity of carbon monoxide (DLCO). Statistical analyses were performed using statistical software Statistica. *Results*: We found correlation of serotonin level with mRSS (r = 0.388, *p* = 0.038). The highest values of serotonin were documented in patients with refractory DU, but this was not statistically significant. We also found a negative correlation between serotonin and FVC (r = −0.397), although it did not reach the level of significance (*p* = 0.114). *Conclusions*: Our study suggests that levels of serum serotonin could affect the course of skin fibrosis and partially restrictive pulmonary dysfunction in patients with SSc. We assume that serotonin might have influence on several features of SSc, but more studies are needed to reveal those relations.

## 1. Introduction

Systemic sclerosis (SSc) is chronic disease characterized by vasculopathy, excessive fibrosis, and production of autoantibodies. Pathological constriction of small vessels causes tissue inflammation that leads to collagen deposition [1,2]. Recently described intra- and extracellular pathways revealed profibrotic roles of numerous cytokines and other molecules [3,4,5], but mechanisms that are in the background of the pathological vasoconstriction in SSc remain obscure. Local hypoxia could be a cause, but also a consequence of pathological vasoconstriction. This process is supported by autoantibody production leading to progressive endothelial cell damage that occurs before fibrosis in SSc is evident. Consequently, fibrosis causes platelet activation and further production of collagen deposits [6]. An actual scientific challenge in SSc diagnosis and early treatment is to define cytokines, hormones, or others markers, that drive pathological process which results in excessive fibrosis.

Common manifestations of SSc can be a consequence of vasculopathy which leads to local ischemia, fibrosis, or both. Raynaud’s phenomenon (RP) is usually the first clinical sign of SSc [7,8]. It is characterized by reversible vasoconstriction, starting as pallor of fingers, followed by cyanosis, and finally by redness. It is, together with digital ulcers (DU), a dominant clinical manifestation of vasculopathy in SSc. Skin fibrosis is clinically assessed by modified Rodnan skin score (mRSS), higher results are related to more thickened skin [9,10]. Interstitial lung disease and fibrosis in SSc is diagnosed by radiologic methods, such as high resolution computed tomography (HRCT), but pulmonary function tests and levels of lung diffusion capacity of carbon monoxide (DLCO) are used in every day diagnostics and follow up. Especially, a decrease in forced vital capacity (FVC) and DLCO are related to progression of the disease [11].

Serotonin is a biochemical messenger and regulator, synthesized from the essential amino acid L-tryptophan. The physiological role of serotonin is especially important in hemostasis, where it causes local vasoconstriction at sites of injury by direct action on smooth muscle or by emphasizing the effect of other vasoconstrictor agents [12,13]. It is found primarily in platelets, in the central nervous system, and gastrointestinal tract, but can also be found in blood plasma. Serotonin is released during platelet activation upon vascular damage, together with several other platelet-derived molecules. Elevated levels of circulating platelet aggregates, as well as different platelet contributions to the vasoconstriction have been described in SSc [12,14]. In the setting of raised local concentration of serotonin during platelet aggregation, the direct effect of serotonin on vascular smooth muscle is vasoconstriction, particularly if there is endothelial dysfunction or damage [12,13]. As the pathological vasoconstriction is presumed as one of the triggers of SSc, we wanted to explore the possible role of serotonin in the disease. The signaling ability of serotonin is well recognized through influencing a wide variety of systemic physiological functions like stimulating monocytes, lymphocytes, and cytokine secretion so it might also have an immunomodulatory role in SSc.

Our aim was to detect the serum serotonin level in SSc patients and analyze it according to the clinical presentations of SSc. We were focused on skin and lung manifestations, such as RP, DU, mRSS, and pulmonary function tests.

## 2. Materials and Methods

Our research was designed as pilot cross-sectional study. It was performed at Department of Clinical Immunology and Rheumatology, University hospital of Split, Croatia. Data collection lasted from September 2016 to March 2017.

We included SSc patients that were diagnosed according to revised ACR/EULAR Classification criteria from 2013 [15]. Some of the included patients that presented with clinical manifestations of SSc also had laboratory findings or some clinical features typical for polymyositis (PM), systemic lupus erythematosus (SLE), or Sjögren’s syndrome (SS). Patients with antidepressants in therapy, especially selective serotonin re-uptake inhibitors (SSRI) and monoamine oxidase inhibitors, were excluded.

We examined our patients during regular rheumatology visits and explained to them the plan of the study. Patients who signed informed consent were included. We recorded mRSS assessed by two rheumatologists, and the presence of RP or DU. Pulmonary function tests were arranged within a few days and were performed on a Jaeger Master screen/Scope device. Forced expiratory volume in the first second (FEV1), FVC, Tiffeneau index (FEV1/FVC ratio), and DLCO were recorded. Pulmonary function test values were calculated as a percentage of predicted normal values for age and sex. Effects of serotonin on blood vessels and tissues are a consequence of a small fraction that is found free in platelet poor plasma. According to available literature, determination of serotonin levels in platelet poor plasma assays is unreliable, with a wide range of reference values and many pre-analytic factors that could affect final results [16]. After consultation with specialists of laboratory diagnostics and biochemistry, we decided to analyze serotonin levels in sera of our patients. Blood samples for analysis of serum values of serotonin were taken in clot activation tubes. After clotting was complete, blood samples were centrifuged for 10 min at 1800× *g*. Then the serum was separated from cells and stored at −20 °C in Department of Laboratory Diagnostics of University hospital of Split, Croatia, until analysis. The concentrations of serotonin were determined with enzyme-linked immunosorbent assay (ELISA) with commercially available reagents (DIAsource Immunoassays SA, Louvain-la-Neuve, Belgium). Manufacturer declared analytical sensitivity of the test was 5 ng/mL with a linearity range of 40 to 860 ng/mL. Reported intra-assay and inter-assay coefficients of variation were 3.9–5.4% and 6%, respectively.

Statistical analyses were performed using statistical software Statistica for Windows, version 12 (TIBCO Software Inc., Palo Alto, CA, USA). Descriptive statistics were provided using the mean and standard deviation (SD). Serotonin values and spirometry parameters were provided using median and range from minimum to maximum. Correlation analysis was used to calculate the association between serotonin and mRSS, and serotonin and spirometric parameters. A Mann–Whitney U test was used when assessing the difference of serum serotonin between patients with and without DU. A Kruskal–Wallis ANOVA test was used to make a distinction of levels of serotonin between patients with restrictive, opstructive, or normal Tiffeneau index. The significance threshold for *p*-value was established at 5%.

## 3. Results

The study included 29 female patients; demographic characteristics are shown in Table 1. The most common SSc presentation was RP in 27 (93.1%) patients. The mean value of mRSS was 8.3 ± 5.9. Considering other vascular manifestations, eight (27.6%) patients had DU and four patients had pulmonary artery hypertension verified by right heart catheterization. Only 19 patients performed pulmonary function testing, while 10 patients were technically unable to perform this exam due to their physical limitations, e.g., due to microstomia or muscular weakness. We found reduced FVC below 80% in six patients, four patients had restrictive spirometry pattern, one had opstructive, and others had normal. Fibrotic lung changes verified by HRCT were described in 15 (51.7%) patients, and in nine (31%) patients DLCO was significantly reduced below 60%. Cumulative results of pulmonary function tests are presented in Table 2. We had 10 patients with laboratory parameters of inflamatory rheumatic disease other than SSc: five with PM, three with SLE, and two with SS. As they first presented as SSc, we decided to include them in our research. Serologic characteristics are shown in Table 1. Considering chronic therapy, almost half of our patients received prednisolon (44.8%) and hydroxychloroquine (44.8%), 27.6% patients were treated with calcium channel inhibitors, and 10.3% with methotrexate.

Median value of serum serotonin was 170 ng/mL (range 24–682 ng/mL). mRSS positively correlated with serum levels of serotonin (r = 0.388, *p* = 0.038), (Figure 1). Although patients with refractory DU had the highest levels of serotonin, we could not perform adequate statistical analysis due to small number of patients with this manifestation (Figure 2). The analysis of spirometry parameters showed a trend of increasing serotonin with decreasing FVC (r = −0.397, *p* = 0.114), (Figure 3). Correlations between serotonin and other spirometry parameters were not statistically significant (for FEV1 r = −0.306, *p* = 0.233, for FEV1/FVC r = 0.169, *p* = 0.516, and for DLCO r = 0.19, *p* = 0.464), (Figure 3). There were no significant correlations considering serologic parameters or therapy regimens.

## 4. Discussion

We showed that higher levels of serotonin correlated with higher mRSS, a clinical marker of skin thickening. This score is used in clinical practice and trials for assessment of different stages of skin fibrosis. In our study, higher levels indicate stimulation of the formation of an extracellular matrix, probably via 5-HT_2B_ receptors [17]. The link of serotonin levels and mRSS, as well as pulmonary function, was occasionally investigated in SSc and our study is one of the first that showed a correlation between levels of serotonin and clinical parameters of skin fibrosis. The majority of studies that examined the effect of serotonin on skin fibrosis were performed on fibroblast cultures or animal models. Dees et al. showed that cultured dermal fibroblasts from SSc patients and healthy individuals respond to serotonin by increased extracellular matrix synthesis [17]. Studies into the role of serotonin in patients with SSc were performed a few decades ago: a British group of authors compared serotonin concentrations in plasma and in platelets in SSc patients and healthy controls and did not find differences between the studied groups [18]. The first clinical reports on the relationship between fibrosis and serotonin date from the 1960s when scleroderma like lesions were observed in patients with metastatic secreting carcinoid tumors [19]. Back in the 1950s, experimental studies revealed that 5-HT could stimulate proliferation of skin fibroblast in rodents resembling skin pathology of SSc [20].

We detected the highest levels of serotonin in patients with DU, although we have not proven a significant relation due to the small sample size. However, as there are limited data on the serotonin effects on DU, we consider these data relevant. It is known that serotonin has a role in different forms of vascular pathology, such as vasospasm of digital arteries, particularly if there is endothelial dysfunction or damage that amplifies platelet aggregation and serotonin releasement. In certain blood vessels, the contractile effects can be markedly enhanced by hypoxia or moderate cooling [21]. A recent study suggested that patients with SSc might have abnormal platelet activation, caused by the damage to blood vessels that characterize the disease [17]. The release of serotonin induced skin fibrosis in experimental animals [17]. Biondi et al. found that plasma serotonin was higher in patients with RP than in healthy controls [22]. Investigation for the role of serotonin in SSc related vasculopathy was continuous and different studies performed genetic analysis of serotonin transporters and receptors, but results were not strong enough for further clinical trials [23,24,25]. Serotonin is an important mediator of bidirectional interactions between the vasoactive amines and the skin which favors and induces fibrosis [26,27]. New findings explain mechanisms of serotonin action on vascular tonus that could cause pulmonary arterial hypertension or RP in SSc [23,28].

A negative correlation between serotonin and FVC is expected, considering the effect on synthesis of extracellular matrix. In the lung tissue, overproduction of extracellular matrix leads to fibrosis, which results in a decrease in FVC and DLCO. Our results suggest that serotonin might have a profibrotic role in lung tissue, especially due to the fact that half of our patients with low FVC had interstitial lung diseases seen by HRCT. There is evidence from mice models that serotonin is involved in pathogenesis of lung fibrosis via 5-HT/Akt signaling pathways and exaggerated TGF-β1 induced collagen synthesis [29,30]. Inhibition of serotonin receptors diminishes progression of bleomycin-induced lung fibrosis in mice [31]. A Swedish group of authors defined the serotonin receptor as a therapeutic target for fibrosing phenotype of interstitial lung diseases [32]. A decrease in of DLCO can be a consequence of both interstitial lung disease and pulmonary arterial hypertension, but as we did not find significant results, we did not perform further analysis. There were no significant correlations between therapy regimens and serum serotonin. It is known that glucocorticoids (GC) have an influence on serotonin signaling pathways in neuropsychiatric diseases and disbalance between GC, and serotonin can induce metabolic syndrome [33,34,35]. Only a few patients in our study received low doses of GC (the highest dose was 7.5 mg of prednisolone per day), so this interaction between GC and serum serotonin was not likely.

The most obvious limitations of our study are the small sample size due to a relatively low incidence of SSc, and the exclusion of patients with the medication that could influence serotonin level. Some clinical exams, such as functional lung tests, are missing in some patients. Another weakness of our study is the determination of serotonin values in serum which consist of serotonin in plasma and serotonin released from platelets. Many pre-analytic and analytic factors can affect results in platelet poor plasma, such as venipuncture procedure, centrifugation process, recovering of serotonin during all phases of sample handling, among others [16]. This study was designed as a pilot study, so we decided to analyze only serum serotonin level. Nevertheless, we found our results promising, and are performing further research with platelet poor plasma together with serum serotonin, which could potentially make a distinction between serotonin released from activated platelets and plasma serotonin.

Fibrosis is the most prominent sign of SSc, and we wanted to stress the importance of the partly neglected role of serotonin, especially in skin thickening. Only a few recent studies have explored this issue. By proving the positive correlation of serotonin and skin changes, as well as changes in pulmonary tests, we presume that our modest results might uncover the small part of SSc enigma. Obvious evidence about the importance of serotonin in SSc can be found in treatment guidelines. The latest European League against Rheumatism (EULAR) recommendations suggest SSRI fluoxetine for treatment of RP, thus implicating the role of serotonin in SSc-related vasculopathy [36]. There is an evident need for more studies on this issue and we will extend our research on larger cohort of SSc patients with more detailed clinical data.

## 5. Conclusions

Our study confirms that serum serotonin level is related to general fibrosis in SSc, especially to skin thickening. Our study also stresses the serotonin influence on DU and, partially, on pulmonary dysfunction in SSc patients.

## Figures and Tables

**Figure 1 medicina-58-00161-f001:**
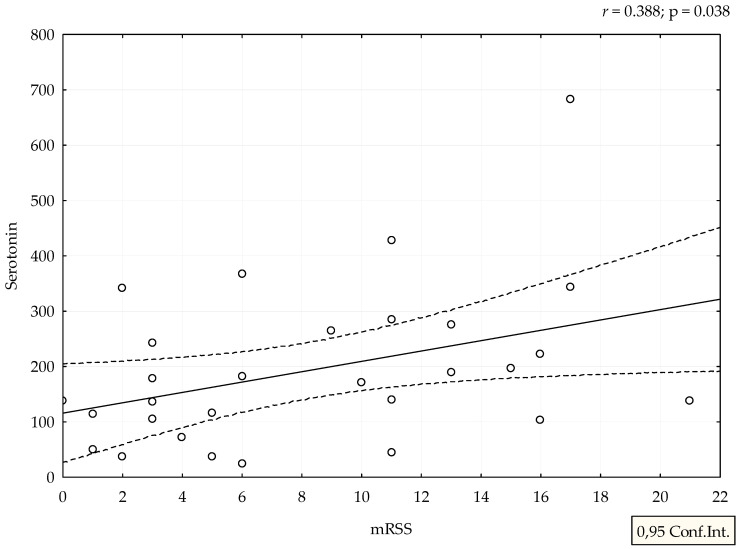
Correlation between mRSS and serotonin levels. Serotonin is expressed as ng/mL.

**Figure 2 medicina-58-00161-f002:**
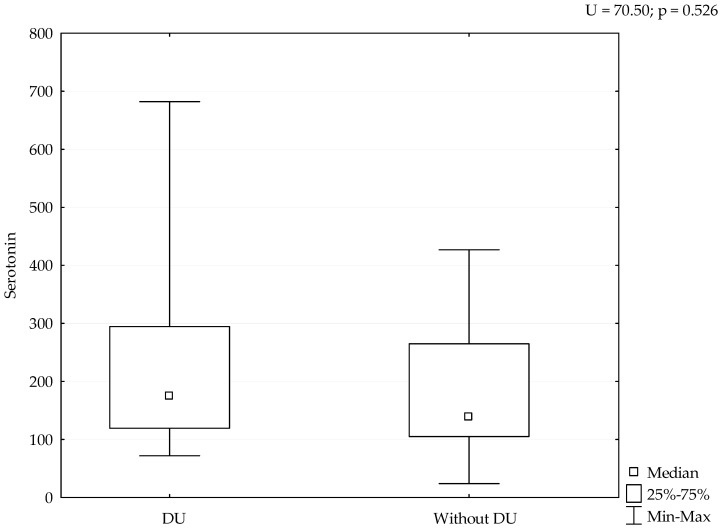
Levels of serotonin considering DU. Serotonin is expressed as ng/mL.

**Figure 3 medicina-58-00161-f003:**
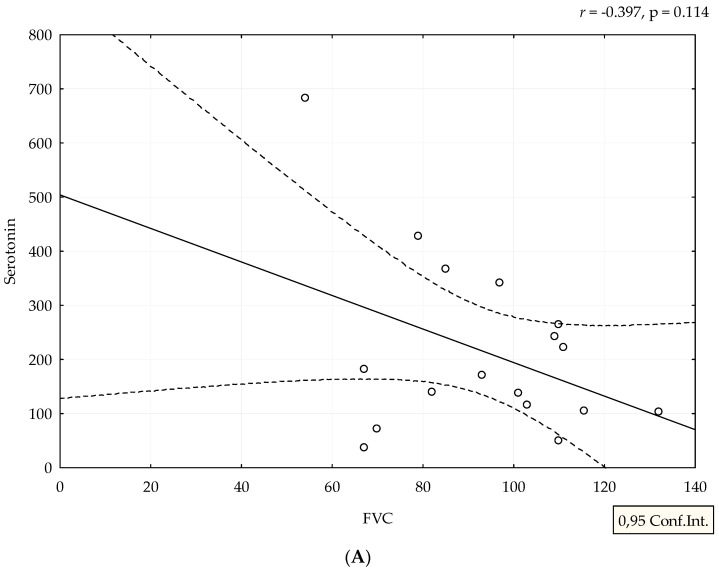
Correlation between spirometry parameters and serotonin levels. Serotonin is expressed as ng/mL. Spirometry parameters (**A**,**B**,**D**) (FVC, FEV1, and DLCO) are expressed as percentage, and (**C**) FEV1/FVC ratio as a number. Figures were created in Statistica 12.

**Table 1 medicina-58-00161-t001:** Demographic characteristics.

	SSc Patients (*n* = 29)
Age (years), median (minimum-maximum)	58 (41–80)
Disease duration (years), mean (S.D. ^1^)	9 (6.4)
Raynoud phenomenon	27 (93.1)
mRSS ^2^, mean (S.D.)	8.3 (5.9)
DU ^3^	8 (27.6)
Interstitial lung disease verified by HRCT ^4^	15 (51.7)
Pulmonary artery hypertension verified by PCWP ^5^	4 (13.8)
ANA ^6^ positive	26 (89.7)
Anti topo I positive	16 (55.2)
ACA ^7^ positive	3 (10)

The most frequent clinical characteristics and serologic parameters are shown. Data are expressed as *n* (%) unless otherwise noted. ^1^ Standard deviation; ^2^ Modified Rodnan skin score; ^3^ digital ulcers; ^4^ high resolution computed tomography; ^5^ pulmonary capillary wedge pressure; ^6^ antinuclear antibodies; and ^7^ anticentromere antibodies.

**Table 2 medicina-58-00161-t002:** Spirometric characteristics.

	SSc Patients (*n* = 19)
FEV1 ^1^	89 (28–114)
FVC ^2^	93 (54–132)
Tiffeneau index	95 (50–108)
Opstructive pattern, *n* (%)	1 (5.3)
Restrictive pattern, *n* (%)	4 (21.1)
DLCO ^3^	60 (12–114)
Reduced diffusion capacity below 60%, *n* (%)	9 (47.4)

Data are expressed as median (minimum–maximum) unless otherwise noted. FEV1, FVC, and DLCO are expressed in values that represent percentages of predicted normal values for age and sex. ^1^ Forced Expiratory Volume in the first second; ^2^ Forced Vital Capacity; and ^3^ Diffusing capacity of the Lung for carbon monoxide.

## Data Availability

Data available on request due to restrictions e.g., privacy or ethical.

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
