# Peer review of "The Levels of Serum Serotonin Can Be Related to Skin and Pulmonary Manifestations of Systemic Sclerosis"

_medicina, 2022, doi:10.3390/medicina58020161_

Round 1
Reviewer 1 Report
In this manuscript Marin Petric and colleagues, characterized the association of serotonin in systemic sclerosis (SSc) and its clinical features. The authors measured in serum of 30 SSc patients with an average disease duration of 9.7 years. Serotonin levels were correlated such as modified Rodnan skin score (mRSS), digital ulcers (DU) and spirometry parameters - forced expiratory volume in the first second (FEV1), forced vital capacity (FVC) and lung diffusion capacity of carbon monoxide (DLCO). They found a correlation with mRSS and a trend of negative correlation between serotonin and FVC. These findings are promising and may pave the basis for further studies where the role of serotonin can be dissected.
The paper is well written, but I have several minor revisions that are needed to be addressed.
Minor:
- The introduction lacks reference as, for example (not only), the following sentences: “Raynaud's phenomenon (RP) is usually the first clinical sign of SSc.”, “Skin fibrosis is clinically assessed by modified Rodnan skin score (mRSS), higher results are related to more thickened skin.” and “Physiological role of serotonin is especially important in haemostasis, where it causes local vasoconstriction at sites of injury by direct action on smooth muscle or by emphasizing the effect of other vasoconstrictor agents.” Please add appropriate references.
- The authors wrote that patients with refractory DU had the highest level of serotonin and this concept is present in the abstract (lines 21-23), in the results (lines 152-154) and in the discussion (lines 183-185) sections, but this observation is not shown along the manuscript. The authors should include this information (also as supplementary material).
- The title is not appropriate for a research article, and it does not reflect the findings in the manuscript. It should be remodulated accordingly.
- The sentence “Serum serotonin analysis was performed, because platelet poor plasma assays has proven much more unreliable, with a wide range of reference values and many pre-analytic factors that could affect results [11].” Is not completely clear, please rephrase it.
- The table 2 is not properly aligned within the text, please check it.
- In figure 1 the X axis start from a negative value. It is more correct starting from zero since the mRSS cannot assume negative value.
- The writing in the figure 2 (axis labels) are not clear, please increase the font.
- I suggest splitting the figure 2 in four different panels (A, B, C and D).
- I suggest including in each graph the p value and the r value to increase the readability of the figures.
- Please insert the code of the ethics committee in the dedicated section (Institutional Review Board Statement)
Author Response
Dear Reviewer,
Thank you for your valuable comments on our manuscript titled “Is the role of serotonin in systemic sclerosis enough elucidated?”. We have carefully read your remarks and addressed them as suggested. Our changes are marked in Track-changes function. Our files were processed in Microsoft Office Word 2007.
Reviewers’ comments:
In this manuscript Marin Petric and colleagues, characterized the association of serotonin in systemic sclerosis (SSc) and its clinical features. The authors measured in serum of 30 SSc patients with an average disease duration of 9.7 years. Serotonin levels were correlated such as modified Rodnan skin score (mRSS), digital ulcers (DU) and spirometry parameters - forced expiratory volume in the first second (FEV1), forced vital capacity (FVC) and lung diffusion capacity of carbon monoxide (DLCO). They found a correlation with mRSS and a trend of negative correlation between serotonin and FVC. These findings are promising and may pave the basis for further studies where the role of serotonin can be dissected. The paper is well written, but I have several minor revisions that are needed to be addressed.
- The introduction lacks reference as, for example (not only), the following sentences: “Raynaud's phenomenon (RP) is usually the first clinical sign of SSc.”, “Skin fibrosis is clinically assessed by modified Rodnan skin score (mRSS), higher results are related to more thickened skin.” and “Physiological role of serotonin is especially important in haemostasis, where it causes local vasoconstriction at sites of injury by direct action on smooth muscle or by emphasizing the effect of other vasoconstrictor agents.” Please add appropriate references.
Answer: We inserted references 7-10 and 13 (Introduction, Page 2, Lines 13, 16, 17, 25 and 33; References, Page 12, Lines 18-23 and 27-29). Reference numbers were additionaly corrected.
The authors wrote that patients with refractory DU had the highest level of serotonin and this concept is present in the abstract (lines 21-23), in the results (lines 152-154) and in the discussion (lines 183-185) sections, but this observation is not shown along the manuscript. The authors should include this information (also as supplementary material).
Answer: We presented this as observation, due to small number of patients with DU. We inserted Figure 2. (Results, Page 6, Line 4 and Page 7, Lines 1-2). Figure numbers were additionaly corrected.
- The title is not appropriate for a research article, and it does not reflect the findings in the manuscript. It should be remodulated accordingly.
Answer: We put another title, hope You would like it (Page 1, Line 2)
The sentence “Serum serotonin analysis was performed, because platelet poor plasma assays has proven much more unreliable, with a wide range of reference values and many pre-analytic factors that could affect results [11].” Is not completely clear, please rephrase it.
Answer: We put an extra explanation, and remodulated the sentence and the part of manuscript which relates to bioanalysis of serotonin (Matherials and Methods, Page 3, Lines 9-15). Additional explanation of difference between determination of serotonin in serum and platelet poor plasma is stated in Discussion section, in a part about limitations of our study (Discussion, Page 11, Lines 9-14).
The table 2 is not properly aligned within the text, please check it.
Answer: As suggested, we made corrections (Results, Page 4, Lines 22-25 and Page 5, Lines 1-5).
- In figure 1 the X axis start from a negative value. It is more correct starting from zero since the mRSS cannot assume negative value.
Answer: As suggested, we made corrections (Results, Page 6, Lines 1-2).
- The writing in the figure 2 (axis labels) are not clear, please increase the font.
Answer: As suggested, we made corrections (Results, Page 7, Lines 3-5, Page 8, Lines 1-5, Page 9, Lines 1-3). Figure number was additionally corrected.
- I suggest splitting the figure 2 in four different panels (A, B, C and D).
Answer: As suggested, we split Figure in four panels (Results, Page 7, Lines 3-5, Page 8, Lines 1-5, Page 9, Lines 1-3). Figure number was additionally corrected.
- I suggest including in each graph the p value and the r value to increase the readability of the figures.
Answer: As suggested, we made corrections (Results, Pages 6-9).
- Please insert the code of the ethics committee in the dedicated section (Institutional Review Board Statement).
Answer: We inserted the code You asked (Institutional review board statement, Page 11, Line 41).
We hope that our explanations and corrections have adequately addressed your concerns and believe that this paper will be of great interest to the readership of your journal considering the clinical implications. Once again, we would like to thank you for your review and valuable suggestions.
On behalf of all authors
With kind regards,
Marin Petrić, MD
Reviewer 2 Report
The topic of this manuscript is interesting. However, the reviewer feel it needs extensive amendments before it can be accepted.
(1) The limitation of this manuscript is obvious. The sample size is too small.
(2) As there was only one male patient, it would be more appropriate to exclude him and focused on the females. The authors should reanalyze the data.
(3) The IRB approval number was not provided.
(4) The data of range, accuracy and precision of the bioanalysis should be provided.
Author Response
Dear Reviewer,
Thank you for your valuable comments on our manuscript titled “Is the role of serotonin in systemic sclerosis enough elucidated?”. We have carefully read your remarks and addressed them as suggested. Our changes are marked in Track-changes function. Our files were processed in Microsoft Office Word 2007.
Reviewers’ comments:
The topic of this manuscript is interesting. However, the reviewer feel it needs extensive amendments before it can be accepted.
(1) The limitation of this manuscript is obvious. The sample size is too small.
Answer: We apsolutely agree and this was stated in the Discussion where we explained limitations of our study. Systemic sclerosis is not a common disease and our study represents a single centre data, so we were not able to collect more patients for mentioned period. Our research was designed as pilot study, so we plan to do new researches with more patients and more detailed analysis.
(2) As there was only one male patient, it would be more appropriate to exclude him and focused on the females. The authors should reanalyze the data.
Answer: As suggested, we exclude that patient and did new statistical analysis. This exclusion did not have significant impact on our results. New data are presented in the manuscript (Abstract, Page 1, Lines16-18, 22-23 and 26, Materials and Methods, Page 27-28 and Results, Page 3, Lines 38-40, 42, 44-47, Page 4, Lines 3-4, 6-7, 9-16, 23-25, Page 5, Lines 1-5, 24, 27-30). Results are also incorporated in Figures.
(3) The IRB approval number was not provided.
Answer: As You asked, we provided the number (Institutional review board statement, Page 11, Line 41).
(4) The data of range, accuracy and precision of the bioanalysis should be provided.
Answer: After consultation with our colleagues from Department of Laboratory of medical diagnostics and biochemistry, we listed available bioanalysis data provided by manufacturer (Matherials and Methods, Page 3, Lines 24-26).
In general remarks You addressed that all five parts of our study have to be improved (Introduction, Study design, Methods, Results and Conclusions). We added some new references in Introduction, put additional explanation in Methods, and reanalyze our data. We hope that our corrections, considering Yours and other Reviewer's suggestions, improved final impression of our research. In case You have another requirements, we are willing to achieve them.
We hope that our explanations and corrections have adequately addressed your concerns and believe that this paper will be of great interest to the readership of your journal considering the clinical implications. Once again, we would like to thank you for your review and valuable suggestions.
On behalf of all authors
With kind regards,
Marin Petrić, MD
Round 2
Reviewer 2 Report
The manuscript has been improved and appears to be acceptable.